# A novel hybrid framework based on temporal convolution network and transformer for network traffic prediction

**Zhiwei Zhang, Shuhui Gong**◉*, **Zhaoyu Liu, Da Chen**

School of Information Engineering, China University of Geosciences, Beijing, China

* shuhui.gong@cugb.edu.cn

## Abstract

### Background

Accurately predicting mobile network traffic can help mobile network operators allocate resources more rationally and can facilitate stable and fast network services to users. However, due to burstiness and uncertainty, it is difficult to accurately predict network traffic.

### Methodology

Considering the spatio-temporal correlation of network traffic, we proposed a deep-learning model, Convolutional Block Attention Module (CBAM) Spatio-Temporal Convolution Network-Transformer, for time-series prediction based on a CBAM attention mechanism, a Temporal Convolutional Network (TCN), and Transformer with a sparse self-attention mechanism. The model can be used to extract the spatio-temporal features of network traffic for prediction. First, we used the improved TCN for spatial information and added the CBAM attention mechanism, which we named CSTCN. This model dealt with important temporal and spatial features in network traffic. Second, Transformer was used to extract spatio-temporal features based on the sparse self-attention mechanism. The experiments in comparison with the baseline showed that the above work helped significantly to improve the prediction accuracy. We conducted experiments on a real network traffic dataset in the city of Milan.

### Results

The results showed that CSTCN-Transformer reduced the mean square error and the mean average error of prediction results by 65.16%, 64.97%, and 60.26%, and by 51.36%, 53.10%, and 38.24%, respectively, compared to CSTCN, a Long Short-Term Memory network, and Transformer on test sets, which justified the model design in this paper.

**Data Availability Statement:** All relevant data are available, and the DOI is: https://doi.org/10.1038/sdata.2015.55.

**Funding:** This work is supported by the Fundamental Research Funds for the Central

Universities, (Grant No. 2-9-2020-028), National Natural Science Foundation of China (Grant No. 62172373), GHFUND B under Grant ghfund 202107021958. Dr. Shuhui Gong is the recipient of all the funding listed above.

**Competing interests:** The authors have declared that no competing interests exist.

## 1. Introduction

With the rapid development of the Internet, Internet traffic is increasing at an explosive rate. Between 2013 and 2018, mobile network traffic increased 61% [1]. This poses a huge challenge to the physical hardware and protocol performance of mobile operators [1]. To be able to meet this challenge and serve users better, mobile operators need to accurately predict the changes in user demand so they can allocate resources appropriately to ensure service quality.

To that end, many scholars have conducted in-depth research on this issue in recent years. At present, the methods proposed by scholars can be divided into traditional prediction methods and prediction methods based on artificial intelligence algorithms. Traditional methods for predicting network traffic are mainly based on traditional mathematical statistics, which use statistical methods such as mean, variance, and expectation as tools for prediction. The most commonly used algorithms in this category are the Auto-Regressive Moving Average (ARMA) algorithm [2] and the Auto-Regressive Integrated Moving Average (ARIMA) algorithm [3]. Scholars have also published many research results on network traffic prediction based on ARMA and ARIMA algorithms. Tian et al. used the ARMA model to predict the network traffic and optimized the prediction results using an optimization function [4]. This research resolved the issue of frequent base station handovers faced by previous studies, due to inaccurate network traffic predictions. Madan et al. used the ARIMA model to predict computer network traffic and also verified that the ARIMA algorithm works mainly for time series of a linear nature [5]. This paper addressed the unresolved issue of accurately forecasting computer network traffic, accommodating both linear and nonlinear time series data, to improve quality of service (QoS) and reduce costs in data centers. However, these types of mathematical statistics-based methods are linear methods that cannot capture the nonlinear and unstable features in network traffic prediction tasks. However, most time-series forecasting tasks are subject to instabilities, such as sudden fluctuations in network traffic due to holidays and unexpected events.

There have also been many attempts to solve the task of network traffic prediction using machine learning. Jain et al. used eXtreme Gradient Boosting (XGBoost) to predict network traffic, which could provide the best solution by capturing the potential features of network traffic prediction from time-series data [6]. Compared to the previous work, this paper addressed the challenges of forecasting network traffic with irregular and abnormal data patterns as well as small cyclical data by effectively combining the Prophet and XGBoost algorithms. Li et al. presented a novel SSVM model that resolves the previously unaddressed issue of nonstationary video traffic prediction in B5G networks by accurately handling all frame types, outperforming existing models in performance [7]. Such machine-learning algorithm-based methods have transformed the time-series problem of network traffic prediction into a supervised learning problem that can better handle more complex issues, such as the fitting of nonlinear relationships.

Recently, deep-learning algorithms have shown better results on many tasks [8]. Moreover, their ability to extract high-dimensional nonlinear features and handle large-scale data was relatively good [9]. It is their ability to handle high-dimensional nonlinear data that has led to the application of deep learning to time-series prediction tasks. Jiang addressed the previously unexplored issue of comprehensively evaluating deep neural networks for Internet traffic prediction, demonstrating that these networks outperform baseline models, with InceptionTime achieving the lowest prediction error in terms of RMSE and MAE [10]. Wan et al. tackled the previously unresolved challenge of limited sample size in network traffic prediction by introducing a novel LSTM-based neural network model with transfer learning [11]. The proposed model outperforms direct training models, achieving a performance improvement of over

40% in network traffic prediction tasks. Ramakrishnan et al. addressed the previously unexplored challenge of using Recurrent Neural Network (RNN) architectures for network traffic prediction in three domains: volume prediction, packet protocol prediction, and packet distribution prediction. The proposed RNN models, including standard RNN, LSTM, and GRU, significantly outperform standard statistical forecasting models, demonstrating their potential in various network traffic prediction applications [12]. It is not common to use Transformer for predicting network traffic, but there are quite a few works using it for similar tasks, such as predicting traffic flow. Liu et al. proposed the STGHTN for traffic flow forecasting. Compared to previous work, the difficulty of predicting complex spatiotemporal data has been reduced. Experiments confirmed its improved prediction accuracy [13]. Pu et al. introduced the MVSTT network for traffic flow prediction, effectively learning complex spatial-temporal correlations from multiple views, providing a comprehensive understanding of traffic flow dynamics. Compared to the work of predecessors, this model has significantly improved accuracy [14]. Huo et al. developed a hierarchical traffic flow forecasting network by merging LTT and STGC to address limitations in previous GCN-based methods. Compared to previous work, this model captures both short-term and long-term temporal relations in traffic flow data, while mitigating the over-smoothing problem [15].

Over the years, research in natural language processing and deep learning has witnessed substantial progress in sequence-to-sequence (seq2seq) modeling, attention mechanisms, and self-attention. Initially, Sutskever et al. introduced the seq2seq model, LSTM in 2014 [16]. The seq2seq model, composed of an encoder-decoder framework, significantly improved the performance of machine translation tasks. However, the model faced limitations in handling long-range dependencies and capturing complex relationships among words in the input and output sequences. To address these challenges, Bahdanau et al. proposed the attention mechanism in 2014 [17]. The attention mechanism allowed the model to weigh the input tokens differently and focus on relevant information during the decoding process. As a result, the model demonstrated improved performance, particularly in capturing long-range dependencies and handling sentences of varying lengths. Further advancements came with the introduction of self-attention and the Transformer architecture by Vaswani et al. in 2017 [18]. The authors replaced the recurrent neural network (RNN) components in seq2seq models with the self-attention mechanism, which enabled parallel processing and efficient learning of long-range dependencies. The Transformer architecture outperformed previous models in various natural language processing tasks, including machine translation and language understanding, by effectively capturing complex word relationships in input and output sequences. Although Transformer directly discards the traditional Convolutional Neural Network (CNN) and RNN, approach is novel. It makes the model lose the ability to capture local features. However, the Temporal Convolutional Network (TCN) can better extract temporal information features [19], and then the convolutional layer in TCN was extended to three dimensions to extract spatial information features in the data. Therefore, we combined TCN and Transformer with a sparse self-attention mechanism to take advantage of both. First, we used TCN based on the Convolutional Block Attention Module (CBAM) attention mechanism [20] to extract spatio-temporal local features and reduce the size of the data. Second, global spatio-temporal features were extracted using a Transformer with a sparse self-attention mechanism [21]. Finally, high-dimensional nonlinear features were extracted and prediction results were obtained using a Multilayer Perceptron (MLP).

Therefore, we propose a Transformer with a sparse self-attention mechanism and TCN optimized for spatial features to handle the spatio-temporal features of network traffic. We named it CBAM Spatio-temporal Convolution Network-Transformer (CSTCN-Transformer) and used it to predict network traffic. The main contributions of this paper are as follows.

1. The spatio-temporal correlation of network traffic was proved, and the model was reasonably improved and designed for the spatio-temporal correlation of network traffic to better extract features.

2. The CSTCN-Transformer deep-learning model was proposed. The improved TCN model was used to extract spatio-temporal features, and the Transformer with a sparse self-attention mechanism was used to further extract the spatio-temporal dependency of the data.

3. By comparing the results of CSTCN-Transformer with each baseline model, we arrived at the conclusion that CSTCN-Transformer is significantly better than the baseline model, which proved the rationality of the method proposed in this paper.

The remainder of this paper proceeds as follows. A detailed model introduction is given in section 2. Section 3 presents the data set used in this paper, the data preprocessing, the experimental steps, the experimental results, and the discussion. Finally, we also discuss the potential applications of the model proposed in this paper.

## 2. Methodology

In this paper, a deep-learning approach was used to predict network traffic. Based on the proof of spatial correlation of network traffic using the Pearson correlation coefficient, we used a deep-learning model that can extract both temporal and spatial features to extract network traffic features. Moreover, we introduced the CBAM attention mechanism and Transformer with a sparse self-attention mechanism into the model. Therefore, the model can better extract important features of network traffic and ignore invalid information that has little effect on the results. The sparse self-attention mechanism of Transformer can also avoid overfitting.

### 2.1 Correlation analysis on time series based on the Pearson correlation coefficient

The Pearson correlation coefficient, also called the Pearson product-moment correlation coefficient, is used to measure the correlation between two sets of data $X$, $Y$. It is defined as the quotient of the covariance and standard deviation between two variables, and its value is between -1 and 1. It is expressed in the equation as:

$$r = \frac{Cov(X, Y)}{D(X)D(Y)} = \frac{E[(X - E(X))(Y - E(Y))]}{D(X)D(Y)}, \tag{1}$$

where $E(X)$ and $D(X)$ denote the mathematic expectation and standard deviation of the variable $X$ respectively, and $Cov(X, Y)$ denotes the covariance of $X$ and $Y$. Substituting the formula for the covariance and standard deviation, the Pearson correlation coefficient $r$ can be expressed as:

$$r = \frac{\sum_{i=1}^{n}(X_i - \overline{X})(Y_i - \overline{Y})}{\sqrt{\sum_{i=1}^{n}(X_i - \overline{X})^2}\sqrt{\sum_{i=1}^{n}(Y_i - \overline{Y})^2}}, \tag{2}$$

where $\overline{X}$ denotes the average value of $X$. It has been proved in the research that the network traffic of the region and the surrounding network traffic have a strong correlation in terms of time series [22]. After confirming that the data exhibits a linear relationship with minimal outliers and follows a continuous normal distribution, we used the most suitable Pearson correlation analysis for this data to investigate the correlation between network traffic data from different regions. By calculating the correlation between the network traffic of region No. 4456

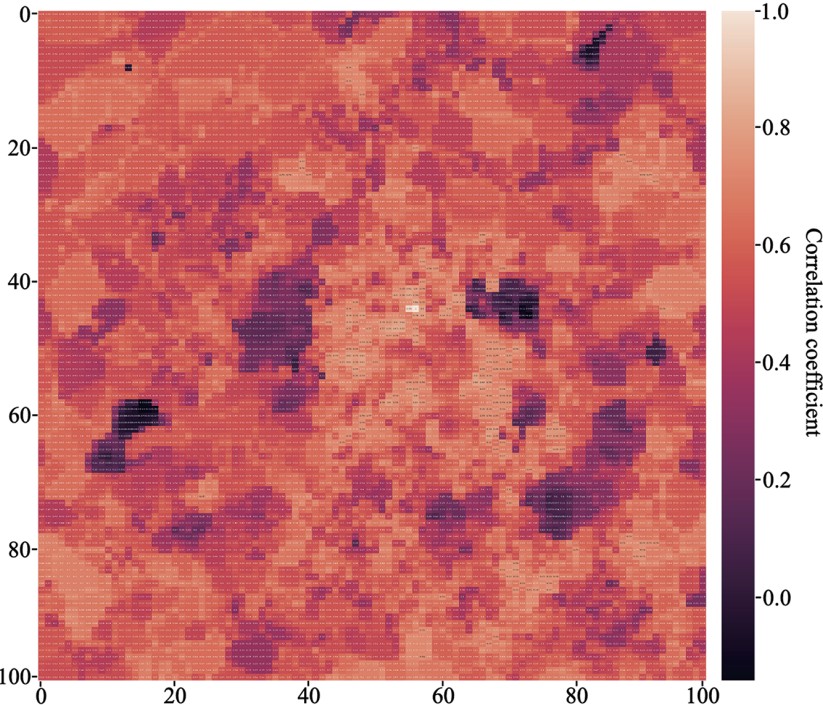

**Fig 1. Pearson heat map of the network traffic flow data of the Milan city network.**

and other regions, we found that the average value of the Pearson correlation coefficient between this region and other regions was just above 0.6, indicating a strong correlation. The specific Pearson heat map is shown in Fig 1.

## 2.2 Deep-learning model

**2.2.1 Brief introduction.**   The proposed method CSTCN-Transformer consists of Transformer with a sparse self-attention mechanism and CSTCN. CSTCN is based on CBAM and TCN. The basis of CBAM and TCN is CNN. Transformer with a sparse self-attention mechanism is based on the original Transformer, as shown in Fig 2. The rest of this section introduces each of these modules in turn.

**2.2.2 CNNs.**   CNNs are deep-learning models that use convolutional kernels to perform convolutional operations on matrices and are commonly used in the field of computer vision for image feature extraction [23]. The original matrix is extracted as a feature matrix with a

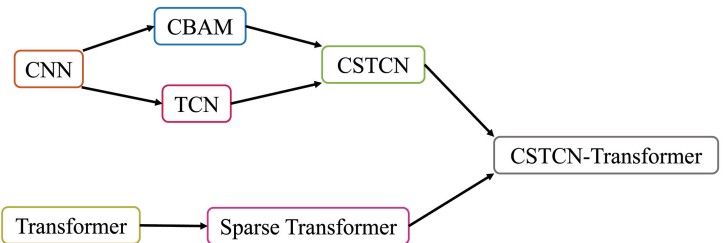

**Fig 2. Relationship between models.**

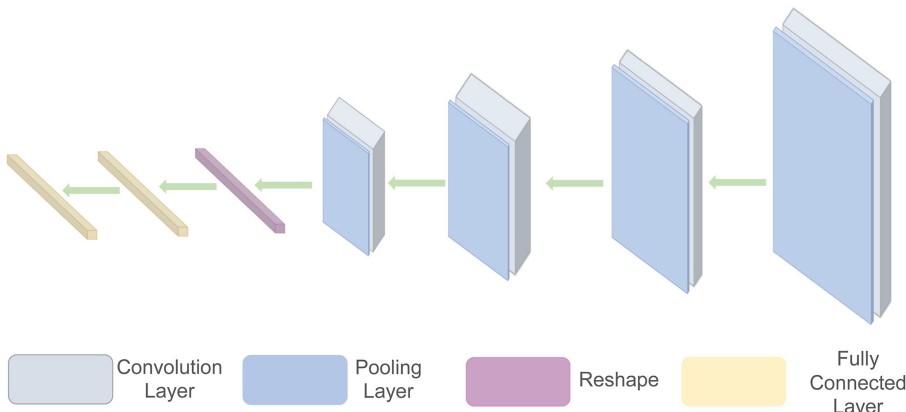

**Fig 3. Schematic diagram of a CNN.**

convolution kernel. It has characteristics such as less loss of spatial information, fewer parameters, and a low training cost. Because there are fewer parameters, the possibility of overfitting is greatly reduced. CNNs include a convolutional layer, a pooling layer, an activation function layer, a fully connected layer, etc. Fig 3 shows a classical CNN.

**2.2.3 STCN.** At present, most TCNs use time-series data for prediction tasks. However, the task in this paper involves the extraction of spatial features. We verified the spatial features of the network traffic data using Pearson correlation coefficients in section 2.1. Therefore, we propose an STCN model that combines 3D CNN and TCN that can extract features in both time and space.

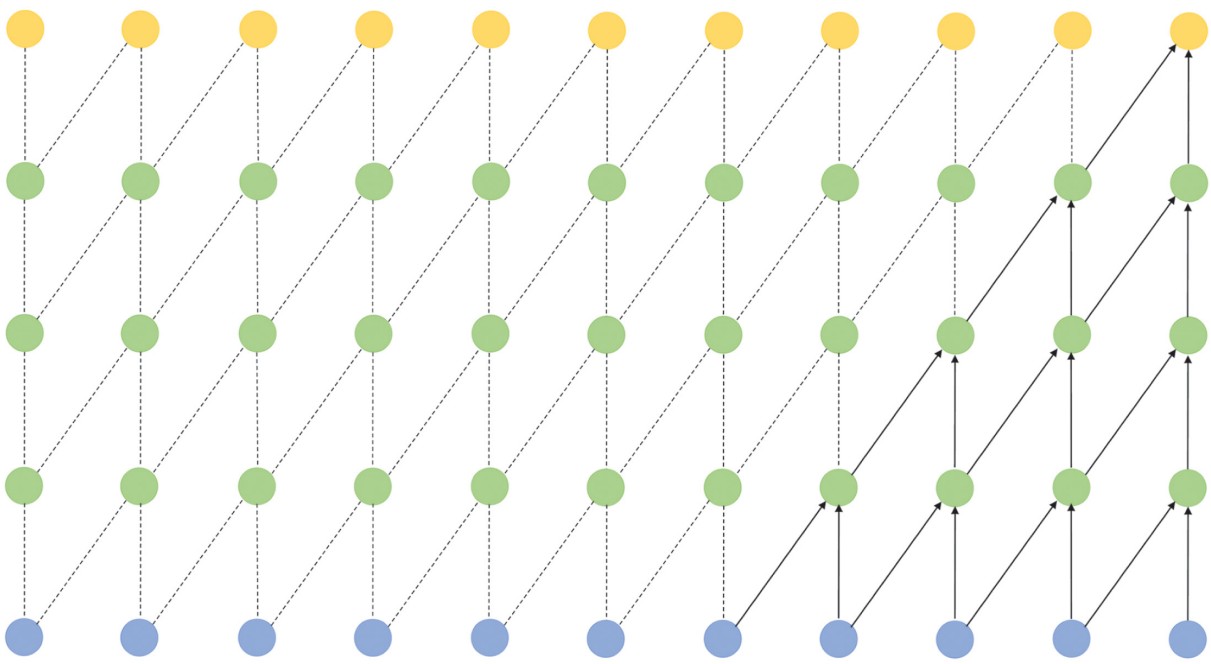

**Fig 4. Schematic diagram of the causal convolution structure.**

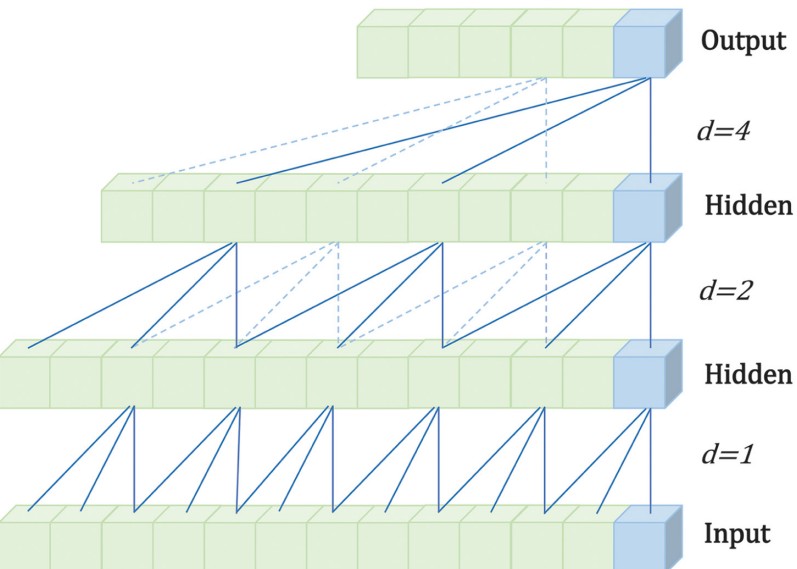

**Fig 5. Schematic diagram of the expansion convolution structure.**

The layers of a TCN have a causal relationship with each other [19]. It can use this causality to maintain the order of data on the time axis. As shown in Fig 4, when predicting data only historical data was used, and future data was shielded to avoid affecting the results. This is a strict time-constrained model.

However, for the causal convolution field in Fig 4, more layers were generally required. The model required more computing resources if the operation was related, as in Fig 4. Therefore, the dilation convolution was introduced to solve this problem, as shown in Fig 5 in the convolution operation at interval sampling. The coefficient $d$ controlled the sampling rate. However, as the number of layers increased, the expansion coefficient delete increased exponentially. Thus, a large sensory field could be obtained without many layers, avoiding the gradient explosion and gradient disappearance caused by too deep a network.

As the number of layers of the STCN model increased, more features were extracted. However, the gradient explosion and gradient disappearance were also caused by the network being too deep. For networks with some depth, the use of residual blocks can be of great benefit in improving the network results [24]. As seen in Fig 6, the input and output were connected using jump connections. This process can be expressed as:

$$H(x) = x + R(x), \tag{3}$$

where $H(x)$ and $x$ denote the output and input of each layer, respectively, and $R(x)$ denotes the residual to be learned. A residual block that used a $1 \times 1$ convolutional kernel was also introduced into the STCN, as shown in Fig 6. A dropout mechanism was also added to the STCN after each causal expansion convolution to prevent overfitting.

The number of convolutional layers required for the spatial features and temporal features was determined by the size of the input region and the length of the input sequence, respectively. It was possible that these two convolutional layers might not match. When these two convolutional layers did not match, there was a problem in processing the data. In this case, an optional convolutional layer with a kernel size of 1×1 could play a role. When the number of convolutional layers required for the temporal features was smaller than the number of

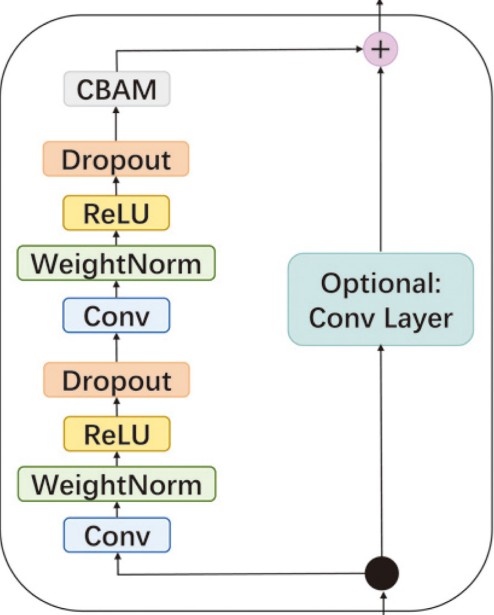

**Fig 6. Schematic diagram of the TCN residual module.**

convolutional layers for the spatial features, the optional convolutional layer with the kernel size and the step size of the time dimension was set to 1. This could prevent the disorder of the time sequence. Conversely, when the number of network layers required for the temporal features was greater than the number of convolutional layers for the spatial features, it was sufficient for the convolutional layers to complete the convolutional computation in the spatial direction after the spatial features were extracted.

**2.2.4 CBAM attention mechanism.** According to psychological principles, animals need to pay attention to noteworthy points effectively in complex environments [25], such as noticing distant water when they are thirsty. The attention mechanism is designed to make the deep-learning model pay attention to some information that has a large impact on the result. When a scene repeatedly appears in the algorithm's field of view, the algorithm will pay more attention to this scene, increase the learning of this part, and avoid the interference of invalid information.

We used the attention mechanism CBAM proposed by Sanghyun Woo et al. in 2018 [20]. As shown in Fig 7, the CBAM block was divided into the Channel Attention Module (CAM) block and the Spatial Attention Module (SAM) block.

As shown in Fig 8, the CAM block was to pass the input feature maps through two parallel MaxPool layers and AvgPool layers, changing the feature maps from the size of C×H×W to the size of C×1×1. It then went through the Share MLP module, and the parameters were shared between these two feature maps. These two feature maps were then added and passed through the sigmoid function. This process can be represented as Eq 4.

$$
\begin{aligned}
M_c(\boldsymbol{F}) &= \sigma(MLP(AvgPool(\boldsymbol{F})) + MLP(MaxPool(\boldsymbol{F}))) \\
&= \sigma(W_1(W_0(\boldsymbol{F}^c_{avg})) + W_1(W_0(\boldsymbol{F}^c_{max}))),
\end{aligned}
\tag{4}
$$

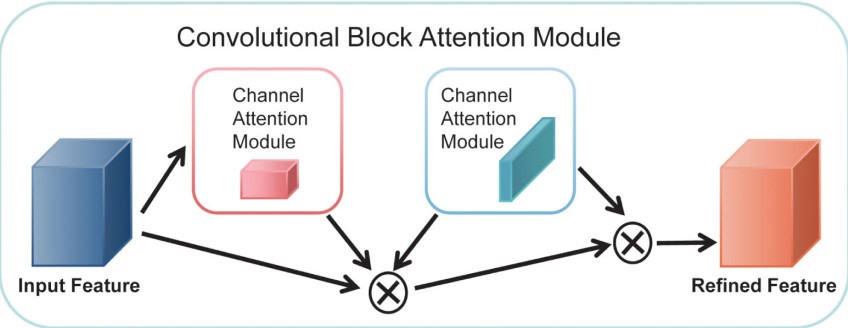

**Fig 7. Schematic diagram of the CBAM block structure.**

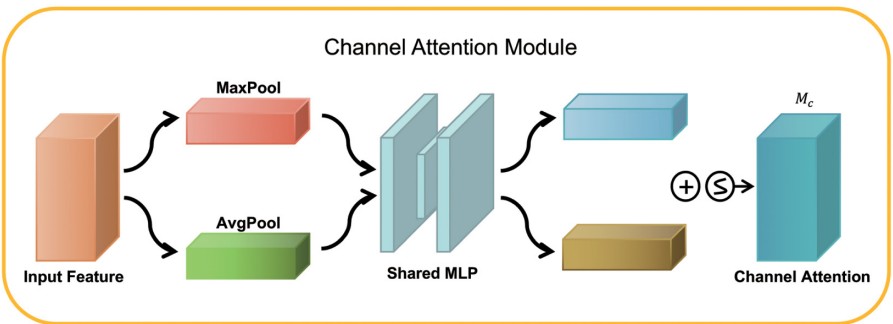

**Fig 8. Schematic diagram of the CAM module structure.**

where $F_{avg}^c$ and $F_{max}^c$ denote average-pooled features and max-pooled features, respectively. $W_0$ and $W_1$ denote the MLP weights. $\sigma$ denotes the sigmoid function.

As in Fig 9, the input feature map of SAM was H×W×C. The SAM block performed maximum pooling and average pooling in one channel dimension to get two H×W×1 feature maps. Next, these two feature maps were spliced together in the channel dimension to become the H×W×2 feature map. This feature map was then passed through a convolution layer and reduced to H×W×1. After that, it passed the sigmoid function. This process can be represented

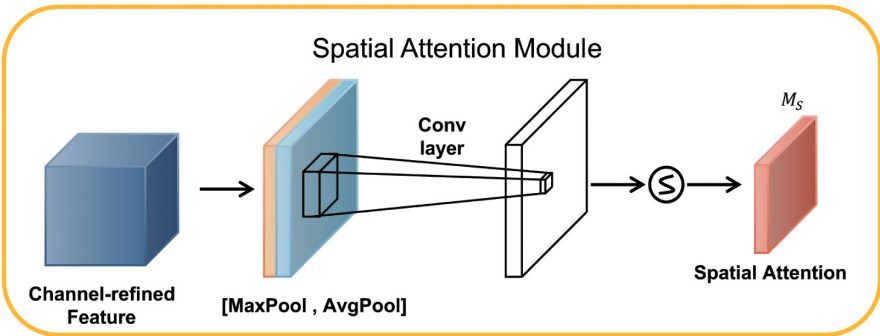

**Fig 9. Schematic diagram of the SAM module structure.**

as Eq 5.

$$M_s(\boldsymbol{F}) \quad = \sigma(f^{7\times7}([AvgPool(\boldsymbol{F}); MaxPool(\boldsymbol{F})]))$$
$$\quad = \sigma(f^{7\times7}([\boldsymbol{F}^s_{avg}; \boldsymbol{F}^s_{max}])),$$

(5)

where $\boldsymbol{F}^s_{avg}$ and $\boldsymbol{F}^s_{avg}$ denote average-pooled features and max-pooled features across the channel, and $f^{7\times7}$ represents a convolution operation with the filter size of 7×7.

**2.2.5 STCN model based on the CBAM attention mechanism.** As mentioned earlier, the attention mechanism can help the algorithm pay more attention to some important information. This means it can pay more attention to those features that have a greater impact on the result and ignore the invalid ones that have almost no impact on the result. Therefore, we introduced the CBAM attention mechanism in the STCN model to help the STCN model extract more important spatio-temporal information, and we named the model CSTCN.

**2.2.6 Transformer with a sparse self-attention mechanism.** Transformer [18] is a deep-learning model based on the self-attention mechanism. In contrast to models such as LSTM and Gated Recurrent Unit (GRU), it allows parallelized processing of data and solves the problem of long-term dependencies in the information transfer process. The Transformer model proposed by Vaswani et al. from Google is used in the field of natural language processing and has recently been used in the field of vision and time-series prediction [26, 27].

As shown in Fig 10, the Transformer as a whole can be divided into two parts, the encoder and the decoder.

The encoder consists of *N* identical network layers, where each network layer consists of a multi-head attention mechanism and Feed-Forward Network, also adding the operation of residual connectivity and normalization. The multi-head attention mechanism was introduced first. Ordinary attention is represented as *output = Attention (Q, K, V)*, the multi-head attention mechanism first copied *Q*, *K*, and *V h* times and then performed the attention interaction operation on each part, with *h* results. The Feed-Forward Network layer is a network layer that provides nonlinear transformations and consists of fully connected layers.

The decoder part also consists of *N* network layers, and the overall structure is similar to that of the encoder. The only difference is that the input to the decoder part came from two sources. In the original Transformer [18], because the problem to be handled was a machine-translation problem, the input of the decoder part had the output of the decoder part corresponding to the *i-1* position in addition to the output of the encoder part. Thus, the inputs of the second attention module, *K* and *V*, came from the encoder, while *Q* was the output of the decoder at the previous position.

The original Transformer used a full self-attention mechanism, as in Fig 11(a). Because the attention scores between each unit were calculated, the memory usage of this attention mechanism was $O(L^2)$, which was not only slower to train but also required more data. Inspired by the work of Li et al. [21], we introduced a sparse self-attention mechanism, as in Fig 11(b). Because there was no longer a need to calculate the attention scores between each unit, the cost of memory usage for each layer became *O(L·log(L))*. This makes the complex Transformer model simpler and more suitable for training time-series prediction tasks with a relatively small amount of data. The work of Li et al. demonstrated [21] that the Transformer with a sparse self-attention mechanism is even better than the Transformer with full self-attention in prediction.

**2.2.7 CSTCN-Transformer.** The deep-learning model CSTCN-Transformer designed in this paper was mainly divided into three parts. The first part was based on the CBAM attention mechanism and the STCN model of the Spatio-Temporal Convolutional Neural Network

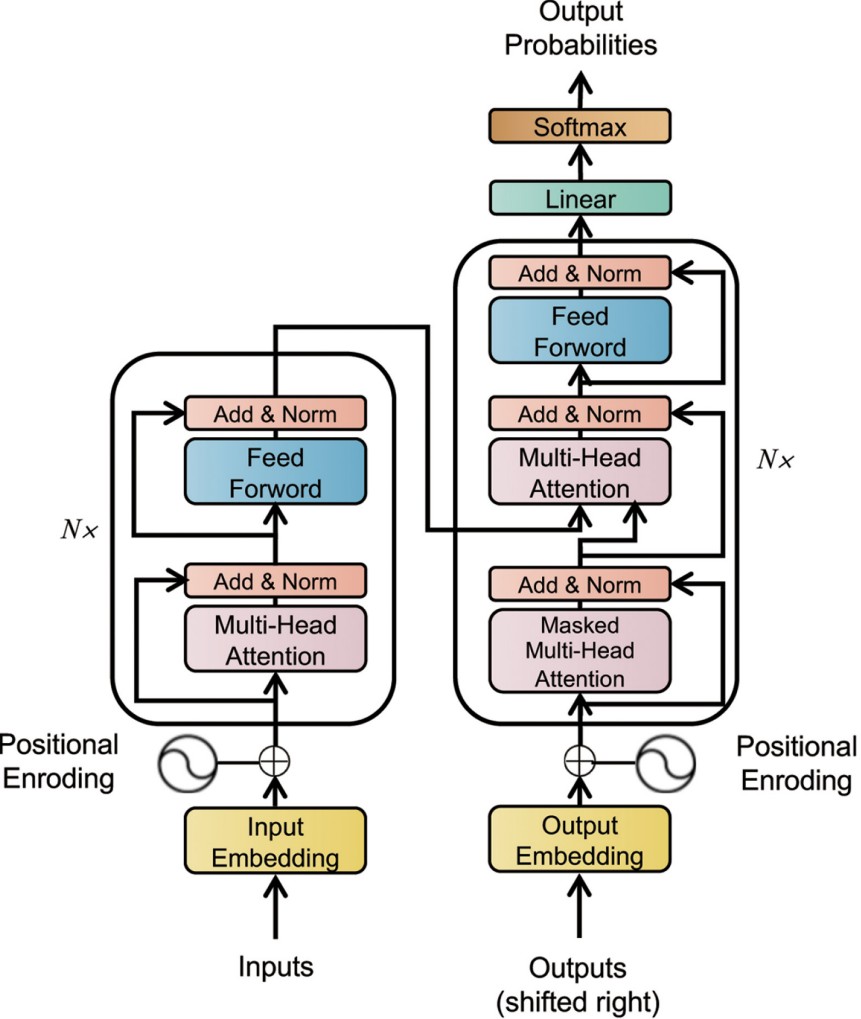

**Fig 10. Schematic diagram of the Transformer structure.**

(CSTCN). It could extract features in both temporal and spatial dimensions and then input these features the second part, the Transformer with a sparse self-attention mechanism. The Transformer model further extracted spatio-temporal features to get the relationship of spatio-temporal features. Finally, the output of the Transformer was extracted by MLP to process the complex high-dimensional nonlinear functional relationships in the feature data, and finally the predicted values were output.

According to the structure of the network shown in Fig 12, it can be seen that the network structure and the extraction of features designed in this paper were analyzed in four main steps.

1. First, a layer of the CNN was used to process the 3D input and get the spatio-temporal features.

2. Next, the spatio-temporal features were further extracted in more detail using the CSTCN deep-learning model.

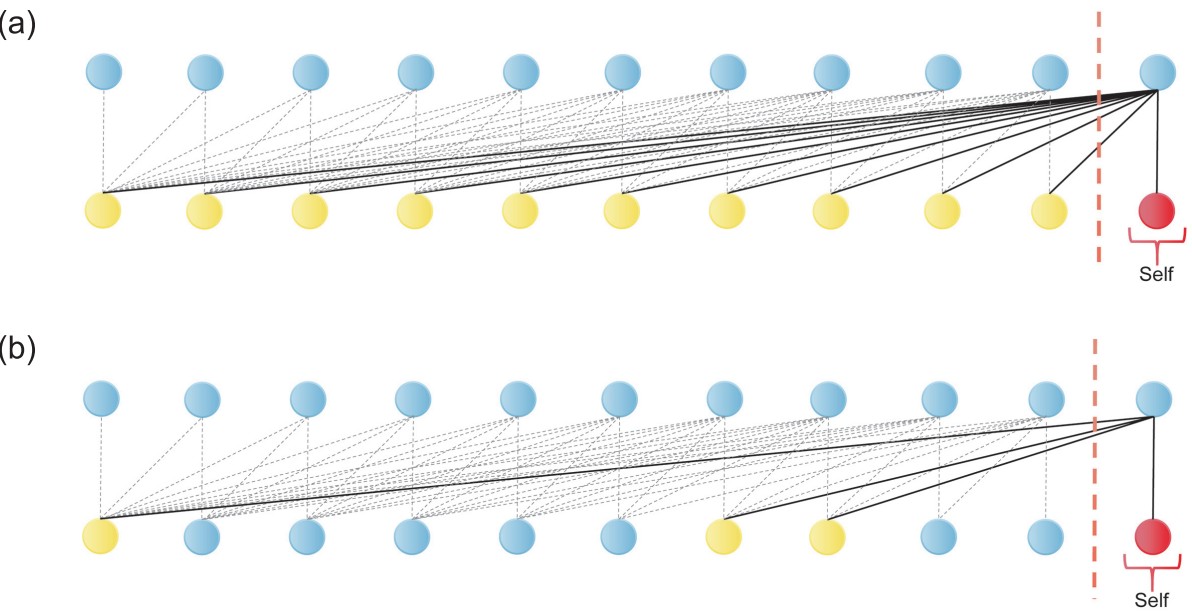

**Fig 11. Schematic diagram of the Transformer self-attention mechanism.**

3. Then the spatio-temporal features were input to the Transformer with a sparse self-attention mechanism in order to obtain its processed spatio-temporal feature data.

4. Finally, the feature data output from Transformer was input to MLP to extract high-dimensional nonlinear features and finally output the prediction results.

In Fig 12, weights and bias refer to the weight and bias values of a fully connected layer or convolutional layer. Variables q, k and v respectively refer to the Query, Key, and Value in the Transformer. The tuple above the arrow indicates the shape of the current data.

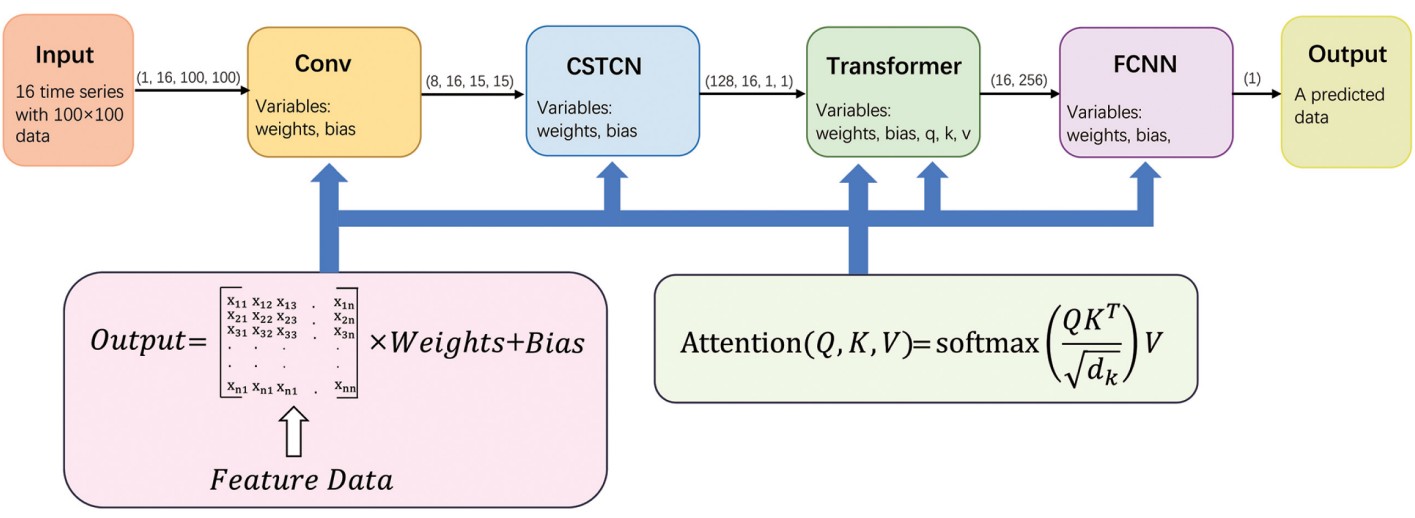

**Fig 12. Overall structure of the CSTCN-Transformer.**

**2.2.8 Loss function.** In this paper, the Log-Cosh regression loss function was chosen for this network, as it is a smoother loss function than L2. Log-Cosh is a logarithm with a hyperbolic cosine. The formula is as follows:

$$L(y, y^p) = \sum_{i=1}^{n} log(cosh(y_i^p - y_i)),$$

(6)

where $y_i$ denotes the ground truth and $y_i^p$ denotes the predicted value. This function is somewhat similar to the mean square error (MSE) formula and has all the advantages of the Huber loss function [28]. It solves the problem that MAE always having large gradients, is also more robust compared to MSE.

# 3. Experiments

## 3.1 Data

### 3.1.1 Introduction of the dataset.

In 2014, Telecom Italia and EIT ICT Lab, SpazioDati, MIT Media Lab, Politecnico di Milano, and other institutions jointly held the "Telecom Italia's Big Data Challenge". The dataset provided by this competition contains telecommunication, weather, news, social network, and electricity data for the city of Milan and the province of Trentino [29]. It can be modeled in several dimensions. In this paper, this dataset was modeled in the spatio-temporal dimension for predicting purposes.

In this paper, we chose mobile network traffic data for the city of Milan. First, we present the form and size of this dataset. Second, we analyze and compare the prediction results of different models with this dataset.

The city of Milan has a total population of about 1.3 million and covers an area of about 552 $km^2$. As shown in Fig 13, this dataset divided Milan into 10,000 grids of 100×100, with each grid table being a realistic size of 235m×235m. Sampling was done every 10 minutes, and

| 9901 | 9902 | $\cdots$ | 9999 | 10000 |
|------|------|----------|------|-------|
| 9801 | $\cdots$ | $\cdots$ | 9899 | 9900 |
| $\cdots$ | $\cdots$ | $\cdots$ | $\cdots$ | $\cdots$ |
| 101 | 102 | $\cdots$ | $\cdots$ | 103 |
| 1 | 2 | 3 | $\cdots$ | 100 |

**Fig 13. Area division map of Milan.**

**Table 1. Spatio-temporal features of dataset.**

| Feature name | Data size | Range | Minimum unit |
|---|---|---|---|
| Temporal features | 4320 | Nov. 1, 2013—Nov. 30, 2013 | Every 10 minutes |
| Spatial features | 100×100 | 10,000 grids of the city of Milan | Network traffic per grid |

there were a total of 4320 samples in the time series, corresponding to the change of mobile network traffic in the city of Milan for a total of 30 days from November 1, 2013 to November 30, 2013.

In summary, the dataset contains both temporal and spatial information and its size is 4320×100×100. Its spatio-temporal features are presented in Table 1.

**3.1.2 Data normalization.** As the skewness of this data was relatively large, we used the log1p method for normalization, which made the data smoother and more consistent with the Gaussian distribution. Next, normalization (Z-score normalization) was performed to make the data more easily trainable. As a linear transformation, normalization would not change the nature of the original data, but would greatly accelerate the gradient descent of the deep-learning algorithm. This process can be expressed using the formula:

$$log1p(x) = ln(1 + x), \tag{7}$$

$$y = \frac{log1p(x) - \mu_{log1p}}{\sigma_{log1p}}, \tag{8}$$

where $\mu_{log1p}$ denotes the mean of $log1p(x)$ and $\sigma_{log1p}$ denotes the variance of $log1p(x)$.

## 3.2 Experimental scheme

**3.2.1 Evaluation criteria.** Performance metrics were necessary steps to evaluate the effectiveness of the prediction model. The purpose of the model in this paper was to predict the network traffic in the next period of time, so the evaluation criterion was to compare the difference between the predicted network traffic and the actual network traffic. The MSE, MAE and the MAPE are commonly used metrics in predicting tasks, and they can be evaluated in different ways to better reflect the performance of the regression model. We used them in this paper to evaluate the predicting accuracy. These three metrics are calculated as follows:

$$MSE = \frac{1}{n}\sum_{i=1}^{n}(\hat{y}_i - y_i)^2, \tag{9}$$

$$MAE = \frac{1}{n}\sum_{i=1}^{n}|\hat{y}_i - y_i|, \tag{10}$$

$$MAPE = \frac{1}{n}\sum_{i=1}^{n}|\frac{\hat{y}_i - y_i}{y_i}|, \tag{11}$$

where $\hat{y}$ is the predicted value and y is the true value.

**3.2.2 Baseline models.** *LSTM*. LSTM is a special kind of RNN network that emerged to solve the problem of long-term data dependence. The timeline of this task was relatively long. As mentioned before, the data used in this paper had 4320 samples in the time series.

Therefore, LSTM was more suitable for this task. We used a layer of LSTM and several fully connected layers to form this network. Recently, Wan et al. employed the model for network traffic prediction [11].

*GRU*. The GRU simplifies the LSTM architecture by merging the forget and input gates into a single update gate, resulting in a more computationally efficient model with comparable performance. Ramakrishnan et al. conducted a study on the performance of GRU for predicting network traffic [12].

*InceptionTime*. InceptionTime is a time series classification approach that incorporates Inception modules in a 1D convolutional neural network. The modules apply parallel convolutional kernels of different sizes, enabling the extraction of features at multiple scales and complexities within the time series data. Jiang investigated the performance of InceptionTime for predicting network traffic [10].

*ResNet*. ResNet, short for Residual Network, is a deep learning architecture that employs residual connections, or skip connections, to bypass layers in the network. These connections facilitate the training of very deep networks by addressing the vanishing gradient problem and improving gradient flow, leading to enhanced performance in a wide range of tasks. Jiang investigated the performance of the model when applied to network traffic prediction tasks [10].

**3.2.3 Ablation experiment models.** *TCN*. TCN is an inflated causal convolution process commonly used for time-series prediction tasks, and it worked better than RNN on many tasks. We also used a TCN and several fully connected layers to form this network.

*CBTCN*. The CBAM block was added to the back of the TCN to enable the model to focus more on processing more important information. Several fully connected layers were added from the end to form this network.

*CSTCN*. The convolutional layers of TCN were extended using a 3D CNN to make it more suitable for processing spatial features. Next, the CBAM block was added to enhance its ability to extract important information.

*Transformer*. Transformer contained a full self-attention mechanism that could better handle spatio-temporal information. In this paper, a layer of Transformer with a full self-attention mechanism and several fully connected layers were used to form this network.

*Sparse-Transformer*. Transformer contained a sparse self-attention mechanism that could avoid overfitting and obtain better regression results. In this paper, a layer of Transformer with a sparse self-attention mechanism and several fully connected layers were used to form this network.

*Adding redundant information*. We expanded the 100×100 tables to 120×120 using random 408 numbers, and the 16 tables on the time series were evenly inserted with four 120×120 tables of 409 random numbers. We use this method to verify the effectiveness of attention mechanisms in reducing the impact of redundant information. For the purpose of achieving the comparative purpose and controlling variables, we use the CSTCN-Transformer to process the data.

**3.2.4 Parameter settings.** The experiments were based on a computer with an RTX3090 graphic processing unit. We used 16 tables of 100×100 before the prediction as the input to the model to predict the present data. In this paper, we used a batch size of 128, an adam optimizer, a learning rate of 0.001, and a dropout rate of 0.02. We used 80% of the data as the training set and 20% of the data as the testing set. The parameters and FLOPs of this model are 3.19M and 916.88M respectively.

Tables 2–6 show the specific hyperparameters of the model proposed in this paper.

**Table 2. Hyperparameters of CSTCN-Transformer (STCN part).**

| Block | Layer | Kernels | Kernel Size | Stride | Padding | Dilation |
|---|---|---|---|---|---|---|
| STCN1 | Conv1 | 16 | (3,3,3) | (1,2,2) | (0,1,1) | (1,1,1) |
|  | Conv2 | 16 | (3,3,3) | (1,1,1) | (0,1,1) | (1,1,1) |
|  | Optional Conv | 16 | (1,1,1) | (1,2,2) | (0,0,0) | (1,1,1) |
| STCN2 | Conv1 | 32 | (3,3,3) | (1,2,2) | (4,1,1) | (2,1,1) |
|  | Conv2 | 32 | (3,3,3) | (1,1,1) | (4,1,1) | (2,1,1) |
|  | Optional Conv | 32 | (1,1,1) | (1,2,2) | (0,0,0) | (1,1,1) |
| STCN3 | Conv1 | 64 | (3,3,3) | (1,2,2) | (6,1,1) | (3,1,1) |
|  | Conv2 | 64 | (3,3,3) | (1,1,1) | (6,1,1) | (3,1,1) |
|  | Optional Conv | 64 | (1,1,1) | (1,2,2) | (0,0,0) | (1,1,1) |
| STCN4 | Conv1 | 128 | (3,3,3) | (1,2,2) | (10,1,1) | (5,1,1) |
|  | Conv2 | 128 | (3,3,3) | (1,1,1) | (10,1,1) | (5,1,1) |
|  | Optional Conv | 128 | (1,1,1) | (1,2,2) | (0,0,0) | (1,1,1) |

## 3.3 Results and analysis

We built the CSTCN-Transformer deep-learning model and trained the model using the training set. To verify the effectiveness of the proposed model, we tested it on the testing set. Fig 14 shows the testing results of the model, whose prediction results maintain a high degree of consistency with the ground truth. In addition, the model can predict the mutation situation better.

We also compared CSTCN-Transformer with several baseline models. The average MSE, MAE and MAPE were obtained for multiple data. The results are shown in Table 7.

We use the percentage calculation method to assess the evaluation criteria for model prediction accuracy with the following equation:

$$p = \frac{b - a}{b} \times 100\%, \tag{12}$$

where $b$ is larger than $a$ and $p$ denotes the percentage decrease of $a$ relative to $b$.

The Fig 15 and Table 7 show that the CSTCN-Transformer significantly outperforms other models in terms of MSE, MAE, and MAPE. Compared to LSTM, GRU, InceptionTime, and ResNet, the CSTCN-Transformer reduces MSE, MAE, and MAPE by 64.91%, 60.76%, 51.94%, and 54.34%; 53.10%, 49.60%, 37.62%, and 43.42%; and 39.66%, 37.41%, 35.47%, and 37.87%, respectively, indicating that the combination of Sparse-Transformer, CBAM, and TCN can indeed better extract useful information and achieve more accurate results.

Among LSTM, GRU, InceptionTime, and ResNet, InceptionTime performs better in MSE, MAE, and MAPE, with values of $1.29 \times 10^{-2}$, $1.01 \times 10^{-1}$, and $3.89 \times 10^{-2}$, respectively. Although the performance of these models in network traffic prediction tasks is not significantly different, InceptionTime shows relatively better performance among these models.

**Table 3. Hyperparameters of CSTCN-Transformer (CBAM pooling part).**

| Block |  |  | Layer | Out Channel | Out Size |
|---|---|---|---|---|---|
| CBAM | CAM |  | Average Pooling | Input Channel | (1,1,1) |
|  |  |  | Max Pooling | Input Channel | (1,1,1) |
|  | SAM |  | Average Pooling | 1 | Input Size |
|  |  |  | Max Pooling | 1 | Input Size |

**Table 4. Hyperparameters of CSTCN-Transformer (CBAM convolution part).**

| Block | | Layer | Input Size | Out Size | Kernel Size | Padding |
|---|---|---|---|---|---|---|
| CBAM | CAM | MLP Conv1 | Input Size | Input Size/16 | (1,1,1) | (0,0,0) |
| | | MLP Conv2 | Input Size/16 | Input Size | (1,1,1) | (0,0,0) |
| | SAM | Conv | 2 | 1 | (7,7,7) | (3,3,3) |

In summary, the CSTCN-Transformer demonstrates a significant advantage in mobile network traffic prediction tasks compared to LSTM, GRU, InceptionTime, and ResNet, which can be attributed to its ability to effectively extract spatiotemporal features and avoid overfitting.

By observing the Fig 16 and Table 8, the following conclusion can be obtained:

First, compared to TCN, CBTCN reduced MSE, MAE, and MAPE by 22.04%, 14.62%, and 6.18%, respectively, demonstrating that the introduction of the CBAM attention mechanism can help the model better focus on useful information, thus improving prediction accuracy. Second, in the comparison between Transformer and Sparse-Transformer, Sparse-Transformer decreased MSE, MAE, and MAPE by 19.23%, 23.53%, and 9.16%, respectively, indicating that the sparse self-attention mechanism can enhance the accuracy of network traffic prediction results and prevent overfitting.

Furthermore, compared to CSTCN, the CSTCN-Transformer reduced MSE, MAE, and MAPE by 65.17%, 51.16%, and 38.33%, respectively, suggesting that the Transformer's self-attention mechanism can help the model extract spatiotemporal features more effectively. Lastly, compared to data with redundant information, the CSTCN-Transformer's MSE, MAE, and MAPE for processing normal data decreased by only 6.65%, 3.83%, and 4.92%, further proving the superior effects of CBAM and Transformer attention, even when dealing with a large amount of redundant information.

In the comparison between Sparse-Transformer and CSTCN-Transformer, CSTCN-Transformer exhibited higher prediction accuracy, reducing MSE, MAE, and MAPE by 50.79%, 33.40%, and 29.69%, respectively. This is attributed to the introduction of CSTCN, which combines spatiotemporal convolutional networks and CBAM attention mechanisms, enabling the model to more effectively extract spatiotemporal features and focus on useful information.

As a model aimed at real-time prediction of network traffic, we wanted to ensure its full potential is utilized. Therefore, we also evaluated its long-term prediction performance with a time interval 10 times greater than the original. The results are presented in Table 9.

**Table 5. Hyperparameters of CSTCN-Transformer (MLP part).**

| Block | Layer | Input Size | Out Size |
|---|---|---|---|
| MLP | Linear 1 | 256 | 1024 |
| | Linear 2 | 1024 | 2048 |
| | Linear 3 | 2048 | 64 |
| | Linear 4 | 64 | 1 |

**Table 6. Hyperparameters of CSTCN-Transformer (Transformer part).**

| Block | n | H | $p_{drop}$ |
|---|---|---|---|
| Transformer | 3 | 3 | 0.1 |

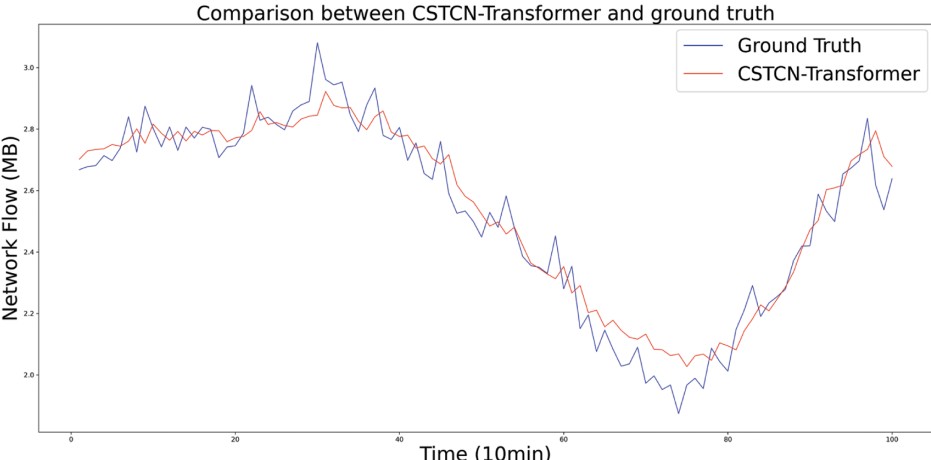

**Fig 14. Prediction effect of CSTCN-Transformer.**

The Table 9 shows that the long-term prediction performance is only slightly worse than the real-time prediction performance. Specifically, the MSE, MAE, and MAPE values for long-term prediction are approximately 9.65%, 6.37%, and 16.73% higher than those for real-time prediction, respectively.

This minor difference between the real-time and long-term prediction performances indicates that the model is quite robust and effective, as it maintains a relatively consistent level of accuracy across varying prediction timeframes.

## 4. Conclusions

We used a deep-learning model combining TCN and Transformer to predict network traffic using the spatio-temporal features of network traffic. Compared with the baseline model, higher prediction accuracy was obtained.

Some of the ablation experiment models used in this paper were modules of the model CSTCN-Transformer proposed in this paper, such as CSTCN and Sparse-Transformer. The baseline models we chose are LSTM and GRU, which are commonly used in this field in recent years. The results of CSTCN-Transformer were more accurate compared to this class of baseline models and ablation experiment models, indicating that each module of the model can be useful in prediction. Other baseline models were other modules that could play the same roles as one of the modules of CSTCN-Transformer; for example, Sparse-Transformer could be replaced by LSTM, GRU or Transformer, and CSTCN could be replaced by CBTCN or TCN. However, the Sparse-Transformer and CSTCN chosen in this paper were significantly higher

**Table 7. Performance comparison with baseline models.**

| Model | MSE | MAE | MAPE |
|---|---|---|---|
| LSTM | $1.77\times10^{-2}$ | $1.34\times10^{-1}$ | $4.16\times10^{-2}$ |
| GRU | $1.58\times10^{-2}$ | $1.25\times10^{-1}$ | $4.01\times10^{-2}$ |
| InceptionTime | $1.29\times10^{-2}$ | $1.01\times10^{-1}$ | $3.89\times10^{-2}$ |
| ResNet | $1.36\times10^{-2}$ | $1.11\times10^{-1}$ | $4.04\times10^{-2}$ |
| CSTCN-Transformer | $6.21\times10^{-3}$ | $6.28\times10^{-2}$ | $2.51\times10^{-2}$ |

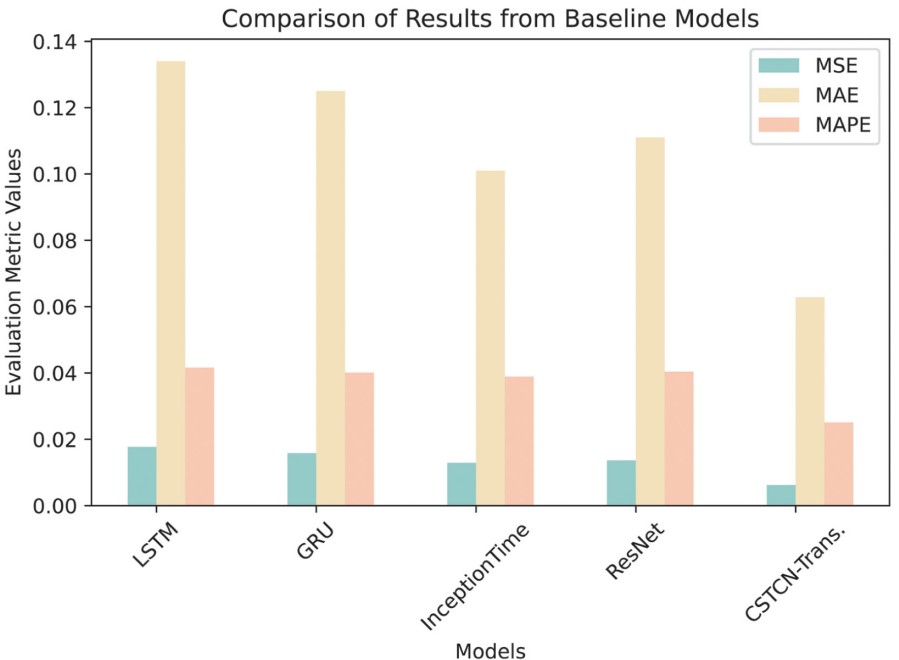

**Fig 15. Comparison of results from baseline models.**

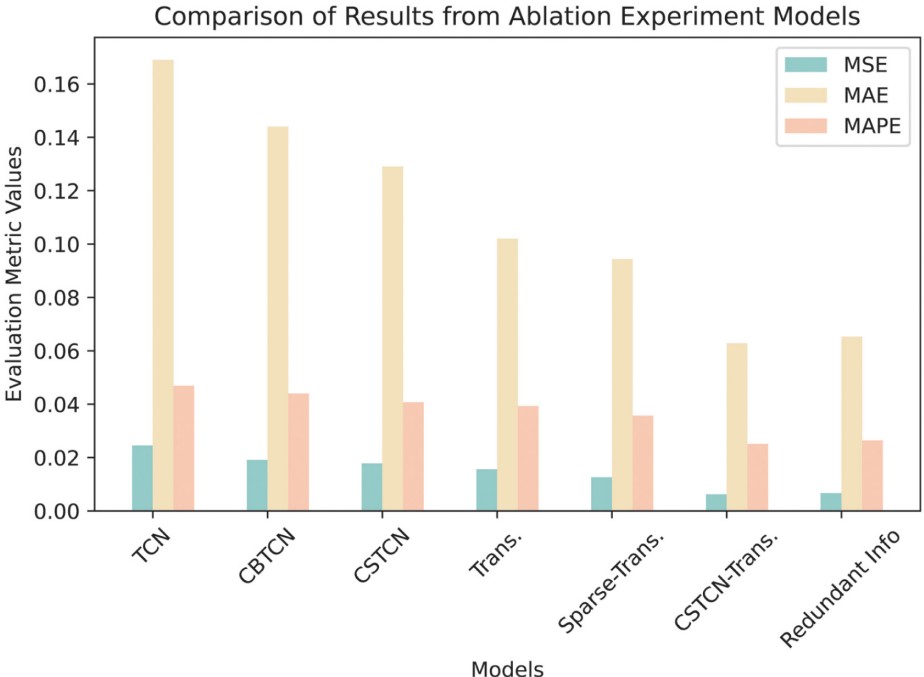

**Fig 16. Comparison of results from ablation experiment models.**

**Table 8. Performance comparison with ablation experiment models.**

| Model | MSE | MAE | MAPE |
|---|---|---|---|
| TCN | $2.45 \times 10^{-2}$ | $1.69 \times 10^{-1}$ | $4.69 \times 10^{-2}$ |
| CBTCN | $1.91 \times 10^{-2}$ | $1.44 \times 10^{-1}$ | $4.40 \times 10^{-2}$ |
| CSTCN | $1.78 \times 10^{-2}$ | $1.29 \times 10^{-1}$ | $4.07 \times 10^{-2}$ |
| Transformer | $1.56 \times 10^{-2}$ | $1.02 \times 10^{-1}$ | $3.93 \times 10^{-2}$ |
| Sparse-Transformer | $1.26 \times 10^{-2}$ | $9.43 \times 10^{-2}$ | $3.57 \times 10^{-2}$ |
| CSTCN-Transformer | $6.21 \times 10^{-3}$ | $6.28 \times 10^{-2}$ | $2.51 \times 10^{-2}$ |
| Adding redundant information | $6.65 \times 10^{-3}$ | $6.53 \times 10^{-2}$ | $2.64 \times 10^{-2}$ |

**Table 9. Comparison of real-time prediction and long-term prediction performance.**

| Model | MSE | MAE | MAPE |
|---|---|---|---|
| Real-time Prediction | $6.21 \times 10^{-3}$ | $6.28 \times 10^{-2}$ | $2.51 \times 10^{-2}$ |
| Long-term Prediction | $6.81 \times 10^{-3}$ | $6.68 \times 10^{-2}$ | $2.93 \times 10^{-2}$ |

than other similar models in terms of prediction accuracy, which proves the rationality of the improved model approach in this paper.

In this paper, we noted the spatial correlation of network traffic, and its spatial features were also considered in the model and extracted. However, the spatial features also had a large amount of redundant information. Thanks to the addition of Transformer and the CBAM block, the model's ability to extract important features was enhanced. It solved the problem of redundant information interference and improved the prediction accuracy. The sparse self-attention mechanism greatly reduced the complexity of the model and solved the overfitting problem, which could handle smaller data sets better.

In terms of practical applications, CSTCN-Transformer could converge to the optimum after about 20 epochs of training, which took about 4 minutes, and the data used in this paper was collected once every 10 minutes, so this model can already be used for real-time network traffic prediction. As shown in Fig 14, this method can effectively deal with irregular and sudden changes of network traffic and get more accurate results. In real life, there is a large amount of data with the same three-dimensional table structure and with similar spatio-temporal features as network traffic, such as electricity and water consumption data. Therefore, the method proposed in this paper can also be applied in the task of predicting such data.

Limitation: The integration of TCN, CBAM, and Sparse-Transformer may lead to high complexity, slowing down training and inference. This could impact performance without a GPU or with limited GPU memory. Hyperparameter tuning: The model involves multiple network structures and numerous hyperparameters. Finding the optimal combination may be time-consuming and resource-intensive.

Our future work will consider using the prediction results of the model in this paper to automatically propose reasonable resource allocation strategies to mobile operators using artificial intelligence algorithms for specific prediction situations. The two components of network traffic predicting and providing strategy are combined to optimize the effectiveness and rationality of operator decisions and to maximize the quality of network services.

## Author Contributions

**Formal analysis:** Zhiwei Zhang.

**Methodology:** Zhiwei Zhang.

**Project administration:** Shuhui Gong.

**Supervision:** Shuhui Gong.

**Visualization:** Zhaoyu Liu, Da Chen.

**Writing – original draft:** Zhiwei Zhang.

**Writing – review & editing:** Shuhui Gong.

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
