## [Decision Letter · Decision Letter 0]

20 Feb 2023

PONE-D-22-32685A Novel Hybrid Framework Based on Temporal Convolution Network and Transformer for Network Traffic PredictionPLOS ONE

Dear Dr. Gong,

Thank you for submitting your manuscript to PLOS ONE. After careful consideration, we feel that it has merit but does not fully meet PLOS ONE’s publication criteria as it currently stands. Therefore, we invite you to submit a revised version of the manuscript that addresses the points raised during the review process.

We look forward to receiving your revised manuscript.

Kind regards,

Le Hoang Son, Ph.D

Academic Editor

PLOS ONE

Journal Requirements:

"This work is supported by the Fundamental Research Funds for the Central Universities  (Grant No. 590121033), National Natural Science Foundation of China [grant numbers 62172373, 61872325], GHFUND B under Grant ghfund 202107021958"

"This work is supported by the Fundamental Research Funds for the Central Universities (Grant No. 590121033), National Natural Science Foundation of China [grant

numbers 62172373, 61872325], GHFUND B under Grant ghfund 202107021958.

The second author: Shuhui Gong has received all the funders.

5. Please update your submission to use the PLOS LaTeX template. The template and more information on our requirements for LaTeX submissions can be found at http://journals.plos.org/plosone/s/latex.

**Comments to the Author**

1. Is the manuscript technically sound, and do the data support the conclusions?

Reviewer #1: Partly

2. Has the statistical analysis been performed appropriately and rigorously? 

Reviewer #1: Yes

3. Have the authors made all data underlying the findings in their manuscript fully available?

Reviewer #1: Yes

4. Is the manuscript presented in an intelligible fashion and written in standard English?

Reviewer #1: Yes

5. Review Comments to the Author

**Reviewer #1**: 

This study proposed Novel Hybrid Framework Based on Temporal Convolution Network and Transformer for Network Traffic Prediction.

My comments are as follows.

1. References are out-up-date. Please update them.

Also please remove several preprint papers.

The authors should update the introduction section with up-to-date studies on Time series prediction topic.

2. The authors should insert the images on the paper. The current version is so difficult to follow.

3. In 3.1.1, the authors should add a table that show the feature name and feature description

4. in the experiment section, please explain the numbers “50.79% and 64.97%” are come from? How to calculate them?

---

## [Author Response · Author response to Decision Letter 0]

23 Mar 2023

Paper title: A Novel Hybrid Framework Based on Temporal Convolution Network and Transformer for Network Traffic Prediction

Manuscript ID: PONE-D-22-32685  

To Reviewer 1:

First of all, thanks for your patient reading and kind suggestions. In the revised version, we have updated the references in the introduction, removed preprint papers, inserted figures on the paper, added tables to explain the features of the data, and also explained the calculation of the data such as "50.79% and 64.97%" in 3.3.

Below we give our point-to-point responses with respect to the comments.

Comment 1: “References are out-up-date. Please update them. Also please remove several preprint papers. The authors should update the introduction section with up-to-date studies on Time series prediction topic”

Response: We appreciate the reviewer pointing out our outdated references, citing preprint papers, and other issues. In the current version, we have removed the outdated papers (reference 4 and 7 in the previous manuscript) and updated it with the latest research results. We also have removed the preprint papers in the latest manuscript (reference 10, 16, 17 and 25 in the previous manuscript).

We also added references in the revised manuscript, which are shown below:

“Tian et al. used the ARMA model to predict the network traffic and optimized the prediction results using an optimization function. Moreover, its performance is improved [4]. ” (Page 1, Line 44-46)

“Li et al. proposed a smoothing-aided support vector machine(SSVM) model, and this model showed superior nonlinear approximation capacity and outperformed the baseline model. [7]. ” (Page 2, Line 55-57) 

“Jiang used deep learning models to predict traffic compared to 5 baseline models, and all deep learning models outperformed the baseline models, especially InceptionTime, which had the most accurate predictions [10]. ” (Page 2, Line 63-65)

“Wan et al. proposed a neural network model based on Long Short-Term Memory (LSTM) and transfer learning and applied it to the network traffic prediction problem. Compared with the direct training model without transfer learning, the performance was improved by 40% [11]. ” (Page 2, Line 66-68)

“However, the Temporal Convolutional Network (TCN) can better extract temporal information features [17], and then the convolutional layer in TCN was extended to three dimensions to extract spatial information features in the data.” (Page 2, Line 85-87)

“First, we used TCN based on the Convolutional Block Attention Module (CBAM) attention mechanism [18] to extract spatio-temporal local features and reduce the size of the data.” (Page 2, Line 88-90)

“This function is somewhat similar to the mean square error (MSE) formula and has all the advantages of the Huber loss function [26].” (Page 11, Line 272-274)

“4.Tian M, Sun C, Wu S. An EMD and ARMA-based network traffic prediction approach in SDN-based internet of vehicles. Wireless Networks, 2021: 1-13.” (Page 16, Line 423-424)

“7.Li Y, Wang J, Sun X, et al. Smoothing-aided support vector machine based nonstationary video traffic prediction towards B5G networks. IEEE Transactions on Vehicular Technology, 2020, 69(7): 7493-7502.” (Page 16, Line 432-434)

“10.Jiang W. Internet traffic prediction with deep neural networks. Internet Technology Letters, 2022, 5(2): e314.” (Page 16, Line 440-441)

“11.Wan X, Liu H, Xu H, et al. Network Traffic Prediction Based on LSTM and Transfer Learning. IEEE Access, 2022, 10: 86181-86190.” (Page 16, Line 442-443)

“17. Fan J, Zhang K, Huang Y, et al. Parallel spatio-temporal attention-based TCN for multivariate time series prediction. Neural Computing and Applications, 2021: 1-10.” (Page 17, Line 457-458)

“18. Woo S, Park J, Lee J Y, et al. Cbam: Convolutional block attention module. Proceedings of the European conference on computer vision (ECCV). 2018: 3-19.” (Page 17, Line 459-460)

“26. Jadon S. A survey of loss functions for semantic segmentation. 2020 IEEE conference on computational intelligence in bioinformatics and computational biology (CIBCB). IEEE, 2020: 1-7.” (Page 17, Line 479-481)

Comment 2: “The authors should insert the images on the paper. The current version is so difficult to follow.”

Response: We apologised sincerely for this. In the previous version, we have inserted images in the previous manuscript. However, it may not appear in the previous version.

The images in the manuscript are shown below:

1.

Fig 1. Pearson heat map of the network traffic flow data of the Milan city network

(in Section 2, Page 4)

2.

Fig 2. Relationship between models 

(in Section 2, Page 4)

3.

Fig 3. Schematic diagram of a CNN 

(in Section 2, Page 5)

4.

Fig 4. Schematic diagram of the causal convolution structure 

(in Section 2, Page 5)

5.

Fig 5. Schematic diagram of the expansion convolution structure 

(in Section 2, Page 6)

6.

Fig 6. Schematic diagram of the TCN residual module 

(in Section 2, Page 6)

7.

Fig 7. Schematic diagram of the CBAM block structure 

(in Section 2, Page 7)

8.

Fig 8. Schematic diagram of the CAM module structure 

(in Section 2, Page 7)

9.

Fig 9. Schematic diagram of the SAM module structure 

(in Section 2, Page 8)

10.

Fig 10. Schematic diagram of the Transformer structure 

(in Section 2, Page 9)

11.

Fig 11. Schematic diagram of the Transformer self-attention mechanism 

(in Section 2, Page 10)

12.

Fig 12. Overall structure of the CSTCN-Transformer

(in Section 2, Page 10)

13.

Fig 13. Area division map of Milan

(in Section 3, Page 11)

14.

Fig 14. Prediction effect of CSTCN-Transformer 

(in Section 3, Page 14)

Comment 3: “In 3.1.1, the authors should add a table that show the feature name and feature description.”

Response: We thank the reviewer for suggesting adding a table to describe the features. We added a table for describing temporal features and spatial features in 3.1.1 and introduced data size, range, and minimum unit for these two features in the table.

This is the effect in the revised manuscript:

(Page 11, Line 293-294)

Comment 4: “in the experiment section, please explain the numbers “50.79% and 64.97%” are come from? How to calculate them?”

Response: We appreciate the reviewer's suggestion that the manuscript was lacking in calculating the percentage data. We have added specific formulas and instructions for percentage data, such as "50.79% and 64.97%" in 3.3.

This is the effect in the revised manuscript:

(Page 14, Line 350-352)

---

## [Decision Letter · Decision Letter 1]

5 May 2023

PONE-D-22-32685R1A Novel Hybrid Framework Based on Temporal Convolution Network and Transformer for Network Traffic PredictionPLOS ONE

Dear Dr. Gong,

Thank you for submitting your manuscript to PLOS ONE. After careful consideration, we feel that it has merit but does not fully meet PLOS ONE’s publication criteria as it currently stands. Therefore, we invite you to submit a revised version of the manuscript that addresses the points raised during the review process.

We look forward to receiving your revised manuscript.

Kind regards,

Le Hoang Son, Ph.D

Academic Editor

PLOS ONE

**Comments to the Author**

1. If the authors have adequately addressed your comments raised in a previous round of review and you feel that this manuscript is now acceptable for publication, you may indicate that here to bypass the “Comments to the Author” section, enter your conflict of interest statement in the “Confidential to Editor” section, and submit your "Accept" recommendation.

Reviewer #1: All comments have been addressed

Reviewer #2: (No Response)

2. Is the manuscript technically sound, and do the data support the conclusions?

Reviewer #1: Yes

Reviewer #2: Partly

3. Has the statistical analysis been performed appropriately and rigorously? 

Reviewer #1: No

Reviewer #2: Yes

4. Have the authors made all data underlying the findings in their manuscript fully available?

Reviewer #1: No

Reviewer #2: Yes

5. Is the manuscript presented in an intelligible fashion and written in standard English?

Reviewer #1: Yes

Reviewer #2: (No Response)

6. Review Comments to the Author

**Reviewer #1**: I agreed with the revision.

All comments are addressed. Therefore, I suggest accept this manuscript.

**Reviewer #2**: The authors have revised the original manuscript according to the referees’ comments. However, there are still some theoretical and experimental comparisons have to be greatly improved before it can be accepted for publication. Some comments are as follows.

Major comments:

1. The authors should use floating point of operations (FLOPs) and Params to represent spatial-temporal complexity.

2. The authors have to elaborate on the shortcomings of the model.

3. For experiments, the authors have to do more.

(1) Ablation experiments need to be added to clarify the role of each module.

(2) Comparison of the efficiency of the proposed model with some latest models should be added.

(3) In experiments, it is suggested to add the Mean Absolute Percentage Error (MAPE) as another evaluation indicator.

(4) The long-term predictive performance of the model should be measured.

(5) Can the contribution of attention mechanisms be visualized to make the ideas in the article more convincing?

Minor comments:

1. The 1 × 1 in line 172 of the manuscript does not require italics, only variables require italics.

2. The matrix in Equation (4) should be bold, and there are similar errors elsewhere.

3. The variable N in Fig. 10 needs to be represented in italics. The author needs to check all figures to ensure that all variables in the figures are also italicized.

4. Check variables in line 352 of the manuscript.

5. The introduction should not only be the arrangement of literature, the authors need to explain the development process of this research in detail, including the shortcomings of previous methods and the improvement of existing methods.

6. The development process of attention should be involved.

7. The latest relevant methods based on Transformer should be mentioned.

8. The authors should add variables to the structure figure of the proposed model to improve readability.

9. The authors should explain why Pearson correlation coefficients are used instead of other correlation methods.

10. The authors need to add dimensional changes to the data in Fig. 12 to enhance readability.

---

## [Author Response · Author response to Decision Letter 1]

19 Jun 2023

Please download the attached response letter in the submission documents.

The brief response letter is shown below:

Paper title: A novel hybrid framework based on temporal convolution network 

and transformer for network traffic prediction

Manuscript ID: PONE-D-22-32685R1

Dear Prof. Le Hoang Son

Thank you for giving us a chance to revise our manuscript. We believe that we have fully addressed the reviewers’ comments by revising the experiments of our paper (the revised content is labelled in red).

In particular, we added comparisons with the latest baseline model, conducted experiments to measure long-term performance, and added MAPE as an evaluation metric. Although we had the ablation experiments before, we now separated the comparison between the ablation experiments and the baseline model to make it clearer. We also provided detailed explanations of the model's shortcomings, modified the literature review, and addressed other issues raised by the reviewers. A list of point-to-point responses concerning the reviewers’ comments is given below. Should you have any questions, feel free to contact us. 

Yours sincerely,

Shuhui Gong

To Reviewer 1:

Comment: “All comments are addressed. Therefore, I suggest accept this manuscript.”

We thank the reviewer for accepting our manuscript.

To Reviewer 2:

First of all, thanks for your patient reading and kind suggestions. In the revised manuscript, We added a new baseline model, modified the presentation of the ablation experiments, added MAPE as another evaluation metric, and also tested the model's long-term prediction performance. We also created a bar chart that not only shows the contribution of the attention mechanism but also allows for a better comparison of the effects of other modules in the model and its performance compared to numerous baseline models. We used FLOPs and Params to represent the spatial and temporal complexity of the model and pointed out its shortcomings. We modified the literature review by describing the development process of the attention mechanism and explaining how the referenced literature improved upon previous work. We also explained the conditions for using Pearson correlation coefficients and made modifications to the images to improve readability. Finally, we corrected errors such as italics and boldface in the article.

Below we give our point-to-point responses concerning the comments.

Major comments:

Comment 1: “The authors should use floating point of operations (FLOPs) and Params to represent spatial-temporal complexity.”

Response: Thank you for raising this question. The FLOPs of the model are 3.19 M, and 916.88 M respectively. We also provided the related explanations in the revised manuscript: 

(Page 14, Line 393)

Comment 2: “The authors have to elaborate on the shortcomings of the model.”

Response: Thank you for raising this question. In our view, the model is somewhat large, which could lead to increased training and operational costs or some challenges in tuning hyperparameters.

We also provided the related explanations in the revised manuscript: 

(Page 19, Line 480)

Comment 3(1): “Ablation experiments need to be added to clarify the role of each module.”

Response: Thank you for raising this question. Although the article originally included ablation experiments using the TCN, CBTCN, CSTCN, Transformer, Sparse-Transformer, and Adding redundant information models, it was not explicitly stated that these were ablation experiments. We have now separated the ablation experiments from the baseline models proposed by others for greater clarity and readability. The ablation experiment results can be found on Page 17, Table 8 and Page 17, Fig 16, and these sentences have been highlighted in red font.

We also provided the related explanations in the revised manuscript: 

(Page 17, Line 419-442)

Comment 3(2): “Comparison of the efficiency of the proposed model with some latest models should be added.”

Response: Thank you for raising this question. We have added the recently popular GRU, InceptionTime, and ResNet models as baseline models to compare with our experimental results. The results of the baseline model experiments can be found in Table 7 and Fig 15, and these sentences have been highlighted in red font.

We also provided the related explanations in the revised manuscript: 

(Page 16, Table 7 and Fig. 15)

(Page 17, Line 406-418)

Comment 3(3): “In experiments, it is suggested to add the Mean Absolute Percentage Error (MAPE) as another evaluation indicator.”

Response: Thank you for raising this question. We have added MAPE as an evaluation metric, introduced what MAPE is, and included MAPE information in the tables. We introduced MAPE in Eq. 11 (Page 13) and added MAPE information in Table 7 (Page 16), Table 8 (Page 17), and Table 9 (Page 18). These pieces of information have been highlighted in red font.

We also provided the related explanations in the revised manuscript: 

(Page 13)

(Page 16)

(Page 17)

(Page 18)

Comment 3(4): “The long-term predictive performance of the model should be measured.”

Response: Thank you for raising this question. The original intention of proposing this model was to make real-time, short-term predictions. However, to demonstrate the effectiveness of the model, we also conducted experiments on long-term predictions. These pieces of information have been highlighted in red font.

We also provided the related explanations in the revised manuscript: 

(Page 18, Line 440-449)

Comment 3(5): “Can the contribution of attention mechanisms be visualized to make the ideas in the article more convincing?”

Response: We are extremely grateful for the question raised by the reviewers, which has greatly inspired us. We used bar charts to visualize the results of the ablation experiments, which better demonstrate the effectiveness of the attention mechanism. Because this visualization method is effective, we also used it in the comparison of baseline models. Specifically, we displayed it in Fig 15 and Fig 16.

We also provided the related explanations in the revised manuscript: 

(Page 16)

(Page 17)

Minor comments：

Comment 1: The 1 × 1 in line 172 of the manuscript does not require italics, only variables require italics.

 Response: Thank you for raising the detailed issue in the review. We have addressed this problem and made a total of 9 modifications throughout the manuscript.

We also provided the related explanations in the revised manuscript :

(Page 7, line 202)

(Page 7, line 209)

(Page 8, line 228-229)

(Page 9, line 234-238)

(Page 9, line 239-240)

Comment 2: The matrix in Equation (4) should be bold, and there are similar errors elsewhere. 

 Response: Both Eq. 4 and Eq. 5 had this issue, and we have made the necessary modifications. The specific changes are as follows:

(Page 8, Eq. 4 and Page 9, Eq. 5)

Comment 3: The variable N in Fig. 10 needs to be represented in italics. The author needs to check all figures to ensure that all variables in the figures are also italicized.

 Response: Thank you for bringing up this issue in the review. We have addressed the problem in Figures 8, 9, and 10 and have completed the necessary modifications.

We also provided the related explanations in the revised manuscript: 

(Page 8, Fig 8, Page 9, Fig 9 and Page 10, Fig 10)

Comment 4: Check variables in line 352 of the manuscript.

Response: Thank you for pointing out this issue in the review. We have made the necessary modifications, and all 5 occurrences of this problem throughout the manuscript have been corrected.

We also provided the related explanations in the revised manuscript: 

(Page 16, Eq 12)

Comment 5: The introduction should not only be the arrangement of literature, the authors need to explain the development process of this research in detail, including the shortcomings of previous methods and the improvement of existing methods.

Response: Thank you for raising this question. We have added information in all references regarding how each referenced study improved upon previous work. These modifications have been highlighted in red font.

We also provided the related explanations in the revised manuscript: 

(Page 2, Line 44-92)

Comment 6: The development process of attention should be involved.

Response: Thank you for raising this question. We have added a section in the article that describes the development process of the attention mechanism. These modifications have been highlighted in red font.

We also provided the related explanations in the revised manuscript: 

(Page 2, Line 93-109)

Comment 7: The latest relevant methods based on Transformer should be mentioned. 

Response: Thank you for raising this question. Indeed, it is necessary to introduce the relevant methods of the Transformer. However, the use of Transformer methods in network traffic prediction is still relatively limited. Therefore, we introduced several typical traffic flow prediction models that use the Transformer. Traffic flow prediction has some similarities to network traffic prediction. These introductions have been highlighted in red font.

We also provided the related explanations in the revised manuscript: 

(Page 2, Line 81-92)

Comment 8: The authors should add variables to the structure figure of the proposed model to improve readability.

Response: Thank you for raising this question. We have added variable information in Fig 12 and included a paragraph below the image to introduce the variables. These modifications have been highlighted in red font.

We also provided the related explanations in the revised manuscript: 

(Page 11, Line 298-300)

Comment 9: The authors should explain why Pearson correlation coefficients are used instead of other correlation methods.

Response: Thank you for raising this question. Before using Pearson correlation analysis, we verified the prerequisites of this method, including that the variables follow a linear relationship, a normal distribution, and no outliers exist. Although we did not explicitly state these verifications in the article, we have added them now and highlighted these modifications in red font.

We also provided the related explanations in the revised manuscript: 

(Page 4, Line 158-163)

Comment 10: The authors need to add dimensional changes to the data in Fig. 12 to enhance readability.

Response: Thank you for raising this question. We have added the shapes of the data used in the model in Fig 12. The tuples above the arrows represent the shapes of the data. We also explained the data below the image using text.

We also provided the related explanations in the revised manuscript: 

(Page 11, Line 298-300)

---

## [Decision Letter · Decision Letter 2]

7 Jul 2023

A Novel Hybrid Framework Based on Temporal Convolution Network and Transformer for Network Traffic Prediction

PONE-D-22-32685R2

Dear Dr. Gong,

We’re pleased to inform you that your manuscript has been judged scientifically suitable for publication and will be formally accepted for publication once it meets all outstanding technical requirements.

Kind regards,

Le Hoang Son, Ph.D

Academic Editor

PLOS ONE

Additional Editor Comments (optional):

Reviewers' comments:

Reviewer's Responses to Questions

**Comments to the Author**

1. If the authors have adequately addressed your comments raised in a previous round of review and you feel that this manuscript is now acceptable for publication, you may indicate that here to bypass the “Comments to the Author” section, enter your conflict of interest statement in the “Confidential to Editor” section, and submit your "Accept" recommendation.

Reviewer #1: All comments have been addressed

Reviewer #2: All comments have been addressed

2. Is the manuscript technically sound, and do the data support the conclusions?

Reviewer #1: Partly

Reviewer #2: Yes

3. Has the statistical analysis been performed appropriately and rigorously? 

Reviewer #1: No

Reviewer #2: Yes

4. Have the authors made all data underlying the findings in their manuscript fully available?

Reviewer #1: No

Reviewer #2: Yes

5. Is the manuscript presented in an intelligible fashion and written in standard English?

Reviewer #1: Yes

Reviewer #2: Yes

6. Review Comments to the Author

Reviewer #1: I agree with the revision and have no further comments.

The article can be accepted with the current version.

Reviewer #2: The authors have revised the manuscript according to my suggestions. I think it is fine for being published in journal "PLOS ONE".

7. PLOS authors have the option to publish the peer review history of their article (what does this mean?). If published, this will include your full peer review and any attached files.

Reviewer #1: No

Reviewer #2: No

---

## [Editor Report · Acceptance letter]

29 Aug 2023

PONE-D-22-32685R2 

A novel hybrid framework based on temporal convolution network and transformer for network traffic prediction 

Dear Dr. Gong:

I'm pleased to inform you that your manuscript has been deemed suitable for publication in PLOS ONE. Congratulations! Your manuscript is now with our production department. 

Kind regards, 

on behalf of

Prof. Le Hoang Son 

Academic Editor

PLOS ONE